# Big data: Airway management at a university hospital over 16 years; a retrospective analysis

Regina Hummel[1☯], Daniel Wollschläger[2☯], Hans-Jürgen Baldering[1☯], Kristin Engelhard[1☯], Eva Wittenmeier[1☯], Katharina Epp[1☯], Nina Pirlich[1☯]*

1 Department of Anaesthesiology, University Medical Centre Mainz, Mainz, Germany, 2 Institute of Medical Biostatistics, Epidemiology and Informatics, University Medical Centre Mainz, Mainz, Germany

☯ These authors contributed equally to this work.
* pirlich@uni-mainz.de

**Data Availability Statement:** All relevant data are within the article and its Supporting Information files.

## Abstract

### Purpose

Little is known about the current practice of airway management in Germany and its development over the last decades. The present study was, therefore, designed to answer the following questions. Which airway management procedures have been performed over the last 16 years and how has the frequency of these procedures changed over time? Is there a relationship between patient characteristics or surgical specialisation and the type of airway management performed?

### Methods

In the present study, we used our in-house data acquisition and accounting system to retrospectively analyse airway management data for all patients who underwent a surgical or medical procedure with anaesthesiological care at our tertiary care facility over the past 16 years. 340,748 airway management procedures were analysed by type of procedure, medical/surgical specialty, and type of device used. Logistic regression was used to identify trends over time.

### Results

Oral intubation was the most common technique over 16 years (65.7%), followed by supraglottic airway devices (18.1%), nasal intubation (7.5%), mask ventilation (1.6%), tracheal cannula (1.3%), double lumen tube (0.7%), and jet ventilation (0.6%). On average, the odds ratio of using supraglottic airway devices increased by 17.0% per year (OR per year = 1.072, 95% CI = 1.071–1.088) while oral intubation rates decreased. In 2005, supraglottic airway devices were used in about 10% of all airway management procedures. Until 2020, this proportion steadily increased by 27%. Frequency of oral intubation on the other hand decreased and was about 75% in 2005 and 53% in 2020.

Over time, second-generation supraglottic airway devices were used more frequently than first-generation supraglottic airway devices. While second-generation devices made

**Funding:** The author(s) received no specific funding for this work.

**Competing interests:** The authors have declared that no competing interests exist.

up about 9% of all supraglottic airway devices in 2010, in 2020 they represented a proportion of 82%.

The use of fibreoptic intubation increased over time in otorhinolaryngology and dental, oral, and maxillofacial surgery, but showed no significant trends over the entire 16-year period.

## Conclusion

Our data represent the first large-scale evaluation of airway management procedures over a long time. There was a significant upward trend in the use of supraglottic airway devices, with an increase in the use of second-generation masks while a decrease in oral intubations was observed.

## Introduction

As failed airway management can have serious impact on patient outcome, the development and optimisation of guidelines is crucial. These airway management guidelines can be further developed and improved based on large clinical data sets. Such datasets are collected, for example, in patient databases such as NAP4 in the UK and the Danish Anaesthesia Database [1,2]. Charlesworth and Agarwal point out in their editorial that secondary analysis of data routinely collected by healthcare providers can help maximise the value of these data and provide meaningful clinical implications [3]. Greenland and Irwin argue that the meaningful use of large-scale patient data is only possible if the variables are mandatorily documented and stored in national databases [4].

In Germany, such a database for airway management does not yet exist, so that little is known about the current practice of airway management in Germany and its development over the last decades [5]. The last assessment of airway management practices at German university hospitals and university-affiliated teaching hospitals was conducted by means of a questionnaire survey after the publication of the first guidelines on airway management in 2004 [6]. In an effort to make the best use of data generated in healthcare facilities, we used routinely collected clinical data from University Medical Centre Mainz in Germany, as suggested by Charlesworth and Agarwal [3]. We performed a retrospective analysis of data collected over the past 16 years to identify changes and trends in airway management over time. Our hypothesis was that airway management has changed during the last two decades in Germany. The research questions underlying the analysis were: Which airway management procedures have been performed over the last 16 years and how has the frequency of these procedures changed over time? Is there a relationship between patient characteristics or surgical specialisation and the type of airway management performed?

Such an analysis may be influenced by confounding factors and temporal biases [3,7,8], but the size of the dataset means that it can still provide information on the frequency and trend of different airway management techniques for specific indications over a long time period and serve as a basis for future adaptation of airway management guidelines.

## Materials and methods

The study was performed in accordance with the Strengthening the Reporting of Observational Studies in Epidemiology (STROBE) statement [9]. Ethical approval for this retrospective

study was deemed not required by the ethical committee due to the absence of identifiable data, as stated in a letter by the local ethical committee (Ethik-Kommission der Landesärzte-kammer Rheinland-Pfalz). Records from the University Medical Centre Mainz internal data acquisition and accounting system (DAQ) of every patient undergoing an operation or medical intervention with anaesthesiologic attendance between January 1, 2005, and December 31, 2020, were screened for airway evaluation and intubation grading data (Mallampati category, neck reclination, thyromental distance, Cormack-Lehane score), airway management techniques (oral intubation, nasal intubation, mask ventilation, supraglottic airway device, tracheal cannula, double lumen intubation, jet ventilation), and patient characteristics (number of patients per specialisation, sex, age, ASA status, type of surgery: elective, urgent, emergency) to generate the final data set. With the intention to report long term trends, data from the beginning of the records were evaluated. Hence, a sample size was not previously determined. Screenshots of the internal DAQ are shown in S1 and S2 Figs.

The DAQ is based on the data set for external quality control in anaesthesia on behalf of the German Society of Anaesthesiology and Intensive Care Medicine (Deutsche Gesellschaft für Anästhesiologie und Intensivmedizin, DGAI) and the Association of German Anaesthesiologists (Bund Deutscher Anästhesisten, BDA) [10]. Datasets can be subdivided by different medical/operative specialties (general surgery; neurosurgery; traumatology; urology; gynaecology; otorhinolaryngology; ophthalmology; dental, oral, and maxillofacial surgery; cardiac surgery; thoracic surgery; orthopaedics; paediatric surgery and medical diagnostics (dermatology, psychiatry, radiology)). Mandatory data input into the DAQ is carried out directly by the responsible anaesthesiologist.

Primary outcome measurements of the study were to analyse the frequency of tracheal intubations or supraglottic airway devices. The full dataset with anonymised patient characteristics was analysed focusing on airway evaluation data (Mallampati category, neck reclination, thyromental distance, Cormack-Lehane score), different airway management techniques (oral intubation, nasal intubation, mask ventilation, supraglottic airway device, tracheal cannula, double lumen intubation, jet ventilation) as well as the use of supraglottic airway devices and fibreoptic intubation in different subspecialties and over a period of 16 years. To differentiate between first- and second-generation supraglottic airway devices, the records of the internal hospital pharmacy were screened for supraglottic airway device purchases in the analysed time. First generation supraglottic airway devices included LMA unique$^{TM}$ and flexible$^{TM}$, AMBU$^®$ AuraFlex$^{TM}$, Aura-i$^{TM}$, AuraOnce$^{TM}$ and air-Q$^®$sp. Second generation devices included LMA Supreme$^{TM}$ and AMBU$^®$ AuraGAIN$^{TM}$.

As further outcomes, demographic patient characteristics (number of patients per specialisation, sex, age, ASA status, type of surgery) were evaluated.

## Statistical analysis

Patient characteristics were summarised using absolute and relative frequencies for categorical variables and means for continuous variables. Binomial logistic regression was used to assess time trends in the proportion of patient characteristics and airway management techniques. A polynomial spline of calendar year was used to model non-monotonic time trends, otherwise a log-linear trend was assumed. Differences in time trends between patient groups were assessed using an interaction term between numerical calendar year and the patient grouping variable of interest. We report adjusted odds ratios together with 95% confidence intervals and p-values from Wald tests based on heteroscedasticity and autocorrelation-consistent standard errors to account for autocorrelation of the time series [11]. The association between the number of supraglottic airway devices as registered in internal pharmacy records and the corresponding

number from the DAQ was evaluated using Poisson regression with only an intercept and log number of masks from DAQ as the offset. The relative rate and 95% confidence interval were calculated using heteroscedasticity and autocorrelation-consistent standard errors. Results of statistical tests were considered significant if the p-value was under 0.05. Data was prepared in Microsoft Excel, analyses were performed using the statistical environment R (version 4.1.2) [12].

## Results

Data from 414,843 patients requiring operation or medical intervention with anaesthesiologic attendance within a period of 16 years were analysed (Table 1).

The mean patient age increased from 44 years in 2005 to 49 years in 2020. The proportion of male patients decreased consistently from 53.3% to 51.8% (OR per year = 0.993, 95% CI = 0.990–0.995. The frequency of ASA status changed over time, with the frequency of ASA III patients slightly increasing and that of ASA I and ASA II patients slightly decreasing. In terms of priority of the operation, elective surgeries showed a slight downward trend over time.

To assess whether airway procedures changed over time, 340,748 airway procedures were analysed by type of procedure (Fig 1).

223,835 of these procedures (65.7%) were performed as oral intubation, 61,558 (18.1%) with *supraglottic airway devices*, 25,574 (7.5%) as nasal intubation, 5,382 (1.6%) with mask ventilation, 4,364 (1.3%) by tracheal cannula, 2,364 (0.7%) with a double lumen tube, and 1,915 (0.6%) by jet ventilation. For 15,756 anaesthesia patients (4.6%) the type of procedure was not evaluated. The proportion of patients anaesthetised with *supraglottic airway devices* showed an upward trend (overall increase of 17.0 percentage points, OR per year = 1.072, 95% CI = 1.071–1.088), while the frequency of oral intubation constantly decreased (overall decrease of 21.9 percentage points, OR per year = 0.936, 95% CI = 0.927–0.944).

In 2005, supraglottic airway devices were used in about 10% of all airway management procedures. Until 2020, this proportion steadily increased about 27%. Frequency of oral intubation decreased to about 75% in 2005 and 53% in 2020).

Fig 2 shows the OR per year for supraglottic airway devices use according to operative/medical specialty and a comparison of time trends of oral intubation and supraglottic airway devices.

The highest OR per year for supraglottic airway device use was observed for anaesthesia in ophthalmology, with a predicted increase of odds of 26.4% per year. When comparing oral intubation with supraglottic airway device use by operative/medical specialty, differences between the specialties were observed both in the general habits of their use and in the development over time (Fig 3).

For example, in neurosurgery, traumatology, and otorhinolaryngology, oral intubation was consistently preferred throughout the entire period, while in ophthalmology, a continuous switch from oral intubation to supraglottic airway device use was obviated.

Data on the type of supraglottic airway devices used for anaesthesia was obtained by analysing the records of the in-house pharmacy for supraglottic airway device purchases, as this was the only way to determine retrospectively the frequency of use for each mask type. The purchase and use of supraglottic airway devices correlated strongly (Pearson's correlation coefficient r = 0.92, p < 0.001, CI 0.77; 0.97). Purchasing records for supraglottic airway devices were 35.2% higher on average than use of supraglottic airway devices as documented in the DAQ. The purchase of second-generation supraglottic airway devices started in 2010. When differentiating the type of supraglottic airway device used by generation, the use of first-

**Table 1. Characteristics of patients for anaesthesia at a tertiary university hospital in Germany between 2005 and 2020 by medical/operative subspecialty, age, sex, ASA status, and priority of operation.** Values are represented as means and relative proportion (%).

| Year | 2005 | 2006 | 2007 | 2008 | 2009 | 2010 | 2011 | 2012 | 2013 | 2014 | 2015 | 2016 | 2017 | 2018 | 2019 | 2020 |
|---|---|---|---|---|---|---|---|---|---|---|---|---|---|---|---|---|
| Patients for anaesthesia | 23,864 | 23,103 | 24,499 | 24,620 | 24,748 | 25,294 | 25,670 | 25,279 | 26,652 | 27,076 | 26,464 | 27,970 | 28,428 | 27,975 | 27,054 | 26,147 |
| **Specialty** | | | | | | | | | | | | | | | | |
| General surgery | 2,041 | 1,929 | 2,055 | 2,426 | 2,462 | 2,337 | 2,574 | 2,714 | 2,808 | 2,819 | 2,524 | 2,692 | 2,417 | 2,286 | 2,114 | 2,358 |
| Neurosurgery | 2,128 | 1,871 | 1,852 | 1,767 | 2,073 | 2,041 | 2,043 | 1,998 | 1,769 | 1,822 | 1,943 | 2,094 | 2,225 | 2,292 | 2,224 | 2,422 |
| Traumatology | 2,891 | 3,001 | 3,174 | 3,027 | 2,994 | 2,955 | 3,141 | 2,715 | 3,424 | 4,279 | 4,106 | 3,941 | 3,933 | 3,858 | 3,579 | 4,148 |
| Urology | 2,772 | 2,739 | 2,926 | 2,727 | 2,842 | 2,815 | 2,832 | 2,828 | 2,784 | 2,785 | 2,621 | 2,760 | 2,823 | 2,898 | 2,775 | 2,681 |
| Gynaecology | 2,504 | 2,432 | 2,643 | 2,799 | 2,794 | 2,977 | 3,382 | 3,369 | 3,235 | 3,205 | 3,238 | 3,763 | 3,910 | 3,787 | 3,799 | 3,369 |
| Otorhinolaryngology | 3,165 | 3.123 | 3.168 | 3,212 | 3,154 | 3,137 | 3,024 | 2,939 | 2,934 | 2,637 | 2,702 | 2,681 | 2,640 | 2,626 | 2,545 | 1,929 |
| Ophthalmology | 1,166 | 1,102 | 1,357 | 1,322 | 1,479 | 1,719 | 1,639 | 1,654 | 2,615 | 2,745 | 2,361 | 2,413 | 2,737 | 2,733 | 2,771 | 2,532 |
| Dental, oral, and maxillofacial surgery | 1,366 | 1,268 | 1,364 | 1,339 | 1,401 | 1,356 | 1,411 | 1,384 | 1,448 | 1,366 | 1,363 | 1,439 | 1,356 | 1,404 | 1,401 | 1,287 |
| Cardiac surgery | 2,217 | 2,186 | 2,479 | 2,617 | 2,646 | 2,656 | 2,563 | 2,806 | 3,169 | 3,421 | 3,479 | 3,634 | 3,653 | 3,364 | 3,101 | 2,672 |
| Medical diagnostics | 368 | 361 | 398 | 506 | 537 | 724 | 900 | 902 | 1,017 | 1,079 | 1,158 | 1,438 | 1,626 | 1,618 | 1,632 | 1,751 |
| Orthopaedics | 1,739 | 1,590 | 1,614 | 1,463 | 1,280 | 1,331 | 1,001 | 886 | 518 | 23 | * | * | * | * | * | * |
| Paediatric surgery | 1,178 | 1,191 | 1,156 | 1,095 | 849 | 985 | 890 | 918 | 892 | 890 | 969 | 1,115 | 1,108 | 1,109 | 1,113 | 998 |
| Thoracic surgery | 329 | 310 | 313 | 320 | 237 | 261 | 270 | 166 | 39 | 5 | * | * | * | * | * | * |
| **Mean age (years)** | 44 | 46 | 46 | 46 | 46 | 47 | 47 | 47 | 47 | 47 | 47 | 47 | 48 | 48 | 48 | 49 |
| **Sex (male), n** | 12,733 | 12,436 | 13,314 | 13,072 | 13,099 | 13,348 | 13,559 | 13,037 | 13,936 | 14,080 | 13,740 | 14,293 | 14,518 | 14,550 | 13,679 | 13,544 |
| **Sex (male), %** | 53 | 54 | 54 | 53 | 53 | 53 | 53 | 52 | 53 | 52 | 52 | 51 | 52 | 52 | 51 | 52 |
| **ASA status** | | | | | | | | | | | | | | | | |
| I, n | 3,395 | 2,952 | 3,006 | 3,018 | 3,421 | 3,378 | 3,498 | 3,608 | 3,776 | 3,345 | 3,210 | 3,319 | 3,401 | 3,687 | 3,446 | 3,166 |
| I, % | 14.2 | 12.8 | 12.3 | 12.3 | 13.8 | 13.4 | 13.6 | 14.3 | 14.2 | 12.4 | 12.2 | 11.9 | 12.1 | 13.1 | 12.9 | 12.1 |
| II, n | 10,461 | 10,260 | 10,963 | 11,094 | 11,323 | 11,927 | 12,076 | 11,662 | 12,161 | 12,390 | 11,485 | 12,098 | 11,928 | 12,055 | 11,489 | 10,876 |
| II, % | 43.8 | 44.4 | 44.7 | 45.0 | 45.8 | 47.2 | 47.1 | 46.2 | 45.8 | 45.8 | 43.6 | 43.5 | 42.4 | 43.1 | 43.0 | 41.6 |
| III, n | 7,354 | 7,162 | 7,725 | 7,592 | 7,192 | 7,227 | 7,556 | 7,659 | 8,290 | 9,030 | 9,047 | 9,425 | 9,547 | 9,310 | 8,998 | 9,125 |
| III, % | 30.8 | 31.0 | 31.5 | 30.8 | 29.1 | 28.6 | 29.5 | 30.3 | 31.2 | 33.4 | 34.3 | 33.9 | 33.9 | 33.3 | 33.7 | 34.9 |
| IV, n | 2,564 | 2,629 | 2,702 | 2,794 | 2,701 | 2,593 | 2,411 | 2,246 | 2,215 | 2,139 | 2,463 | 2,837 | 3,099 | 2,719 | 2,631 | 2,859 |
| IV, % | 10.7 | 11.4 | 11.0 | 11.3 | 11.0 | 10.3 | 9.4 | 8.9 | 8.3 | 7.9 | 9.3 | 10.2 | 11.0 | 9.7 | 9.9 | 11.0 |
| V, n | 82 | 98 | 96 | 112 | 95 | 105 | 96 | 71 | 106 | 115 | 143 | 115 | 140 | 161 | 125 | 113 |
| V, % | 0.3 | 0.4 | 0.4 | 0.5 | 0.4 | 0.4 | 0.4 | 0.3 | 0.4 | 0.4 | 0.5 | 0.4 | 0.5 | 0.6 | 0.5 | 0.4 |
| H, n | 8 | 2 | 7 | 10 | 15 | 19 | 13 | 7 | 3 | 6 | 6 | 5 | 7 | 6 | 11 | 8 |
| H, % | 0.03 | 0.01 | 0.03 | 0.04 | 0.06 | 0.08 | 0.05 | 0.03 | 0.01 | 0.02 | 0.02 | 0.02 | 0.02 | 0.02 | 0.04 | 0.03 |
| **Priority of operation** | | | | | | | | | | | | | | | | |
| elective, n | 19,742 | 18,545 | 19,708 | 19,356 | 19,826 | 20,179 | 20,476 | 19,979 | 21,132 | 21,265 | 20,564 | 21,613 | 21,760 | 21,998 | 20,890 | 19,834 |
| elective, % | 82.8 | 80.3 | 80.4 | 78.6 | 80.1 | 80.0 | 80.0 | 79.1 | 80.0 | 78.7 | 78.0 | 77.7 | 77.4 | 78.7 | 78.2 | 75.9 |
| urgent, n | 2,672 | 2,887 | 3,038 | 3,137 | 2,742 | 2,725 | 2,814 | 2,903 | 3,079 | 3,362 | 3,248 | 3,381 | 3,246 | 3,144 | 3,328 | 3,752 |
| urgent, % | 11.2 | 12.5 | 12.4 | 12.7 | 11.1 | 10.8 | 11.0 | 11.5 | 11.6 | 12.4 | 12.3 | 12.2 | 11.5 | 11.3 | 12.5 | 14.3 |
| emergency, n | 1,450 | 1,671 | 1,753 | 2,127 | 2,180 | 2,345 | 2,360 | 2,371 | 2,340 | 2,398 | 2,542 | 2,805 | 3,116 | 2,796 | 2,482 | 2,561 |
| emergency, % | 6.1 | 7.2 | 7.2 | 8.6 | 8.8 | 9.3 | 9.2 | 9.4 | 8.8 | 8.9 | 9.6 | 10.1 | 11.1 | 10.0 | 9.3 | 9.8 |

*In 2015, the Orthopaedics department was merged with the Traumatology department and the Thoracic Surgery department with the Cardiac Surgery department.

generation supraglottic airway devices decreased, while the use of second-generation supraglottic airway devices increased over time (interaction calendar year by mask generation). (Fig 4). While second-generation devices made up about 9% of all supraglottic airway devices in 2010, 2020 they represented a proportion of 82%.

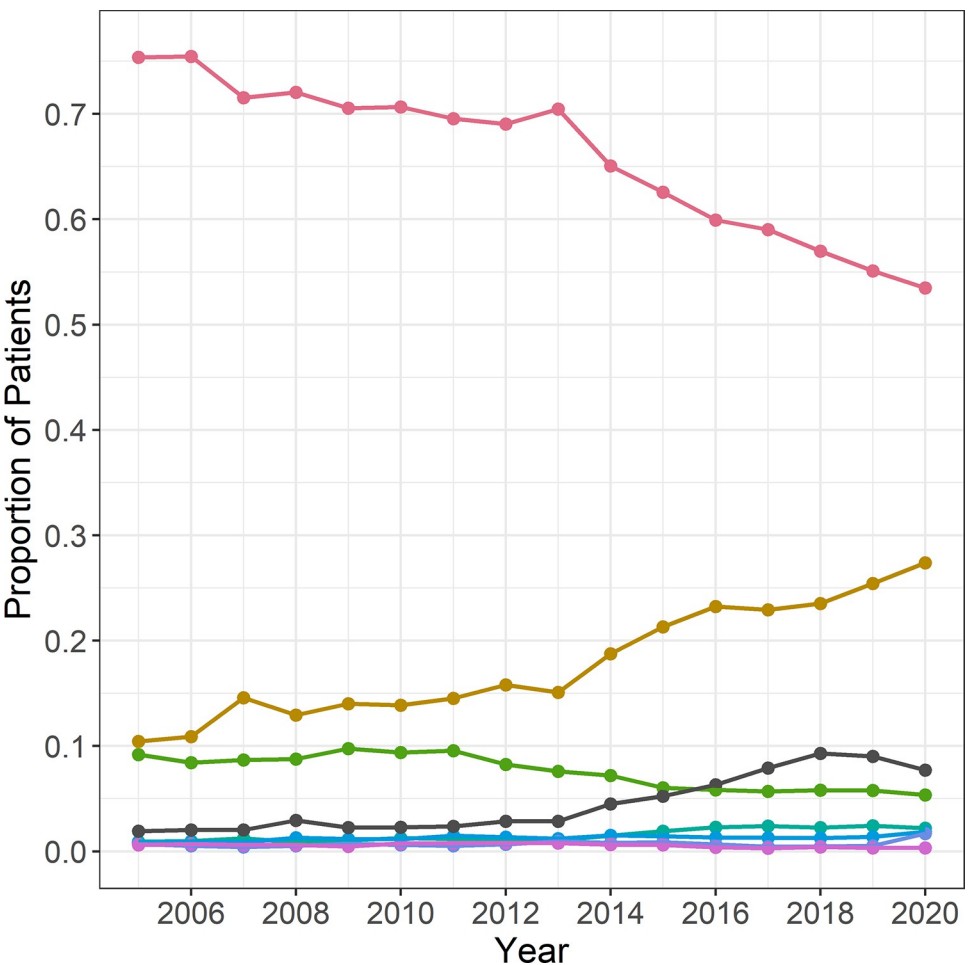

**Fig 1. Types of airway procedures used from 2005 to 2020.** (oral intubation = red, *supraglottic airway device* = brown, nasal intubation = green, mask ventilation = turquoise, tracheal cannula = blue, double lumen intubation = purple, jet ventilation = black).

Mallampati categories exhibited a different trend over time. The frequency of the Mallampati I category decreased from 56.7% to 30.2% while the Mallampati II category increased from 26.6% to 43.9% over the 16. years.

Cormack-Lehane scores II, III, and IV remained stable over the 16-year time, while the frequency of Cormack-Lehane I decreased. The number of unevaluable Cormack-Lehane scores constantly increased over time.

A total of 18,614 fibreoptic intubations were performed (5.5% of all airway management procedures). Otorhinolaryngology as well as dental, oral, and maxillofacial surgery were the specialties with most frequent fibreoptic intubation use (6,873 and 6,361 fibreoptic intubations, respectively). Overall, the numbers of fibreoptic intubation use showed non-linear variations over the years (Fig 5).

## Discussion

The present study is the first large-scale analysis of longitudinal data on procedures used in periinterventional airway management over a 16-year period. This enabled us to assess long-term trends in the use of specific airway management procedures and devices. In the future,

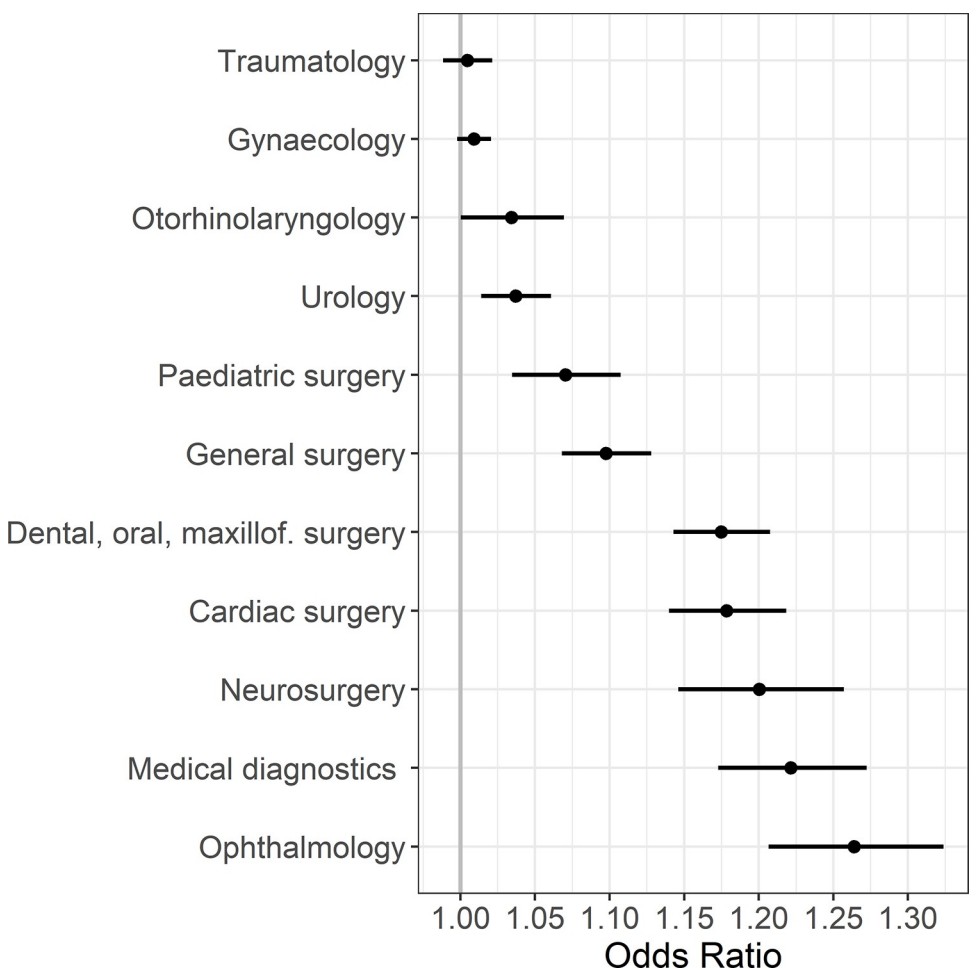

**Fig 2. Odds ratios per year with 95% confidence intervals for supraglottic airway device use according to operative/medical specialty.**

these datasets can serve as a basis for optimizing recommendations for procedures and guidelines for airway management.

The dataset of the present longitudinal-study includes 340,748 airway interventions over a 16-year period. To our knowledge, there are only few studies with a comparable large data set of airway-interventions, which, however, do not cover this vast period of time. In the United Kingdom, a national census of airway management techniques collected 114,904 data from patients undergoing general anesthesia in 309 hospitals over a 2-week period in 2008 [13]. While the dataset of the latter study represents only one point in time in 2008, the present study reflects a longitudinal development over 16 years and has collected about three times as much data. This should be taken into account when comparing these two studies. In the UK study, a supraglottic airway device was used in 56.2% of cases, followed by tracheal tube (38.4%) and facemask (5.3%) [13]. In the present study, the frequency of tracheal intubation decreased from 75.4% initially to 58.5% in 2020, while at the same time the supraglottic airway device was used in only 10.4% of patients in 2005, which increased to 27.4% in 2020. This shows a clear trend that the use of supraglottic airway devices in Germany increased over time, but even in 2020, half as many supraglottic airway devices were used at our hospital as in the UK [14] in 2008. Data from another German university hospital including 167,349 anesthetic

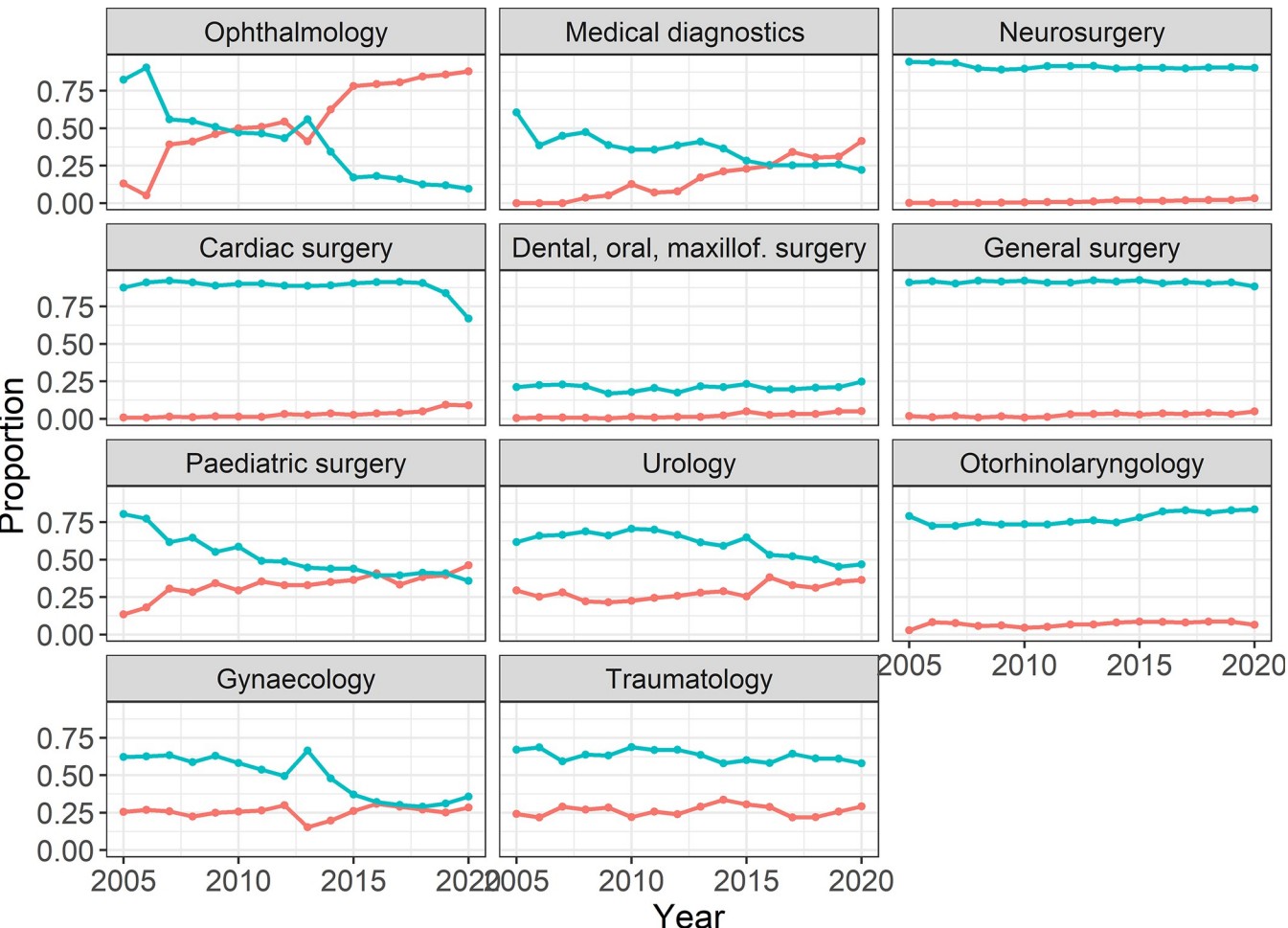

**Fig 3. Supraglottic airway device (red) use over time compared to oral intubation (turquoise) trends over time by operative/medical specialty.**

records over a 6-year period (2005–2011) revealed a proportion of 20.6% for the use of supraglottic airway devices [14]. In comparison, our data resulted in a proportion of 13.1% between 2005 and 2011. One possible reason for the difference between data sets of the two university hospitals in Germany and the data of the UK cross-sectional study could be the more complex surgical procedures in a university setting requiring tracheal intubation. Overall, it must be assumed that the nationally and institutionally differences of supraglottic airway device use seem influenced by an emotional discussion about the "safe" airway (intubation) and the "unsafe" airway (supraglottic airway device). Regardless, any method is only as good as its use and patient safety should always be paramount.

Woodall and Cook´s data showed also that only 10% of supraglottic airway devices used in 2008 were second-generation [13]. In the present analysis, purchase of second-generation supraglottic airway devices started in 2010 and since 2015, more second-generation than first-generation supraglottic airway devices have been applied. The replacement of first-generation with second-generation masks is a desirable trend, as noted by Cook et al. because second-generation masks are superior in regards to their efficacy and safety, particularly for expanded indications [15]. Second-generation supraglottic airway devices are characterised by reduced oropharyngeal leak and the possibility of inserting a gastric tube, which increases patient

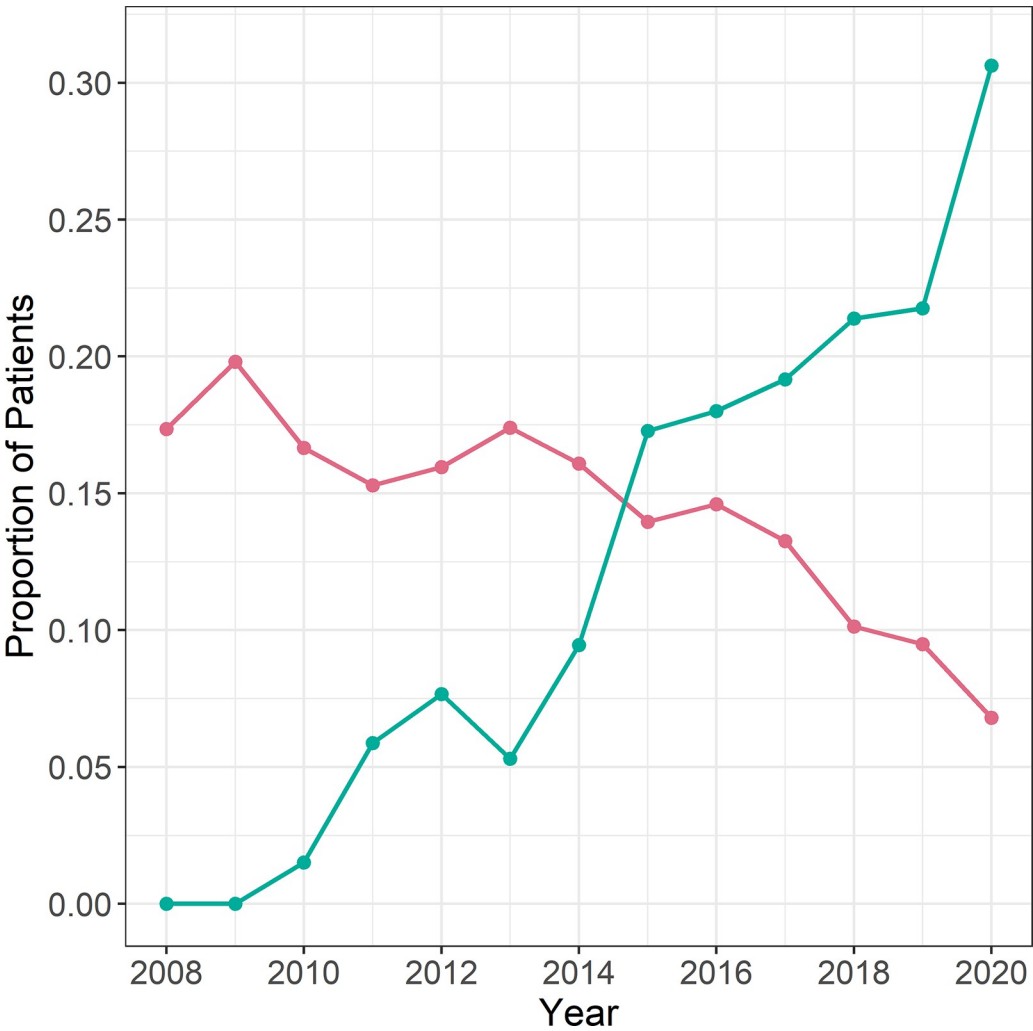

**Fig 4. Purchase of first- (red) and second-generation (turquoise) supraglottic airway devices over time (records of the internal hospital pharmacy).**

safety. The present study was able to confirm that second-generation supraglottic airway devices have gained acceptance over time due to their clear advantages. The observed trend of replacing oral intubation with supraglottic airway devices is particularly evident in operative/ medical specialities that benefit from the expanded indications of second-generation masks. The improved seal and the identification of possible malpositions have led to an increased use in areas with limited access to the airway, such as in ophthalmology or medical diagnostics (e.g., radiology).

Surprisingly, the frequency of Mallampati categories I and II have changed over time. One possible reason for the decrease in the Mallampati category I in favour of Mallampati category II could be that anaesthesiologists are less afraid of managing a difficult airway due to the increased availability of videolaryngoscopy and supraglottic airway devices compared to the past. This made the preoperative assessment of a difficult airway less relevant, which may have led to less familiarity with the score and, in case of doubt, to an overestimation of the Mallampati category. This assumption was confirmed by a survey of European anaesthesiologists on theoretical knowledge and practical skills regarding the Mallampati category. The survey

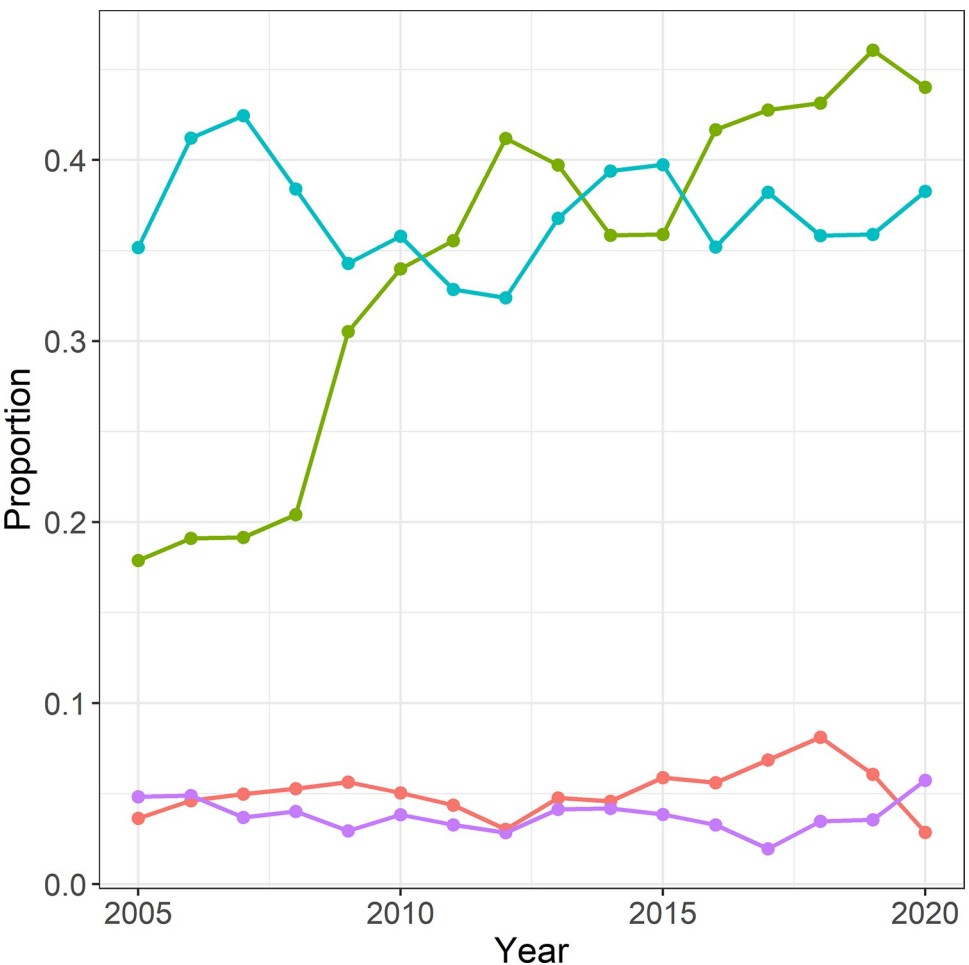

**Fig 5. Relative frequency of fibreoptic intubations by specialty.** (otorhinolaryngology = turquoise, dental, oral, and maxillofacial surgery = green, traumatology = purple, neurosurgery = red).

revealed large knowledge gaps among anaesthetists, which were attributed to the lack of interest or insufficient training in airway assessment [16]. Therefore, in future, anaesthesiologists' training should again place more emphasis on carefully communicating the high value of preoperative assessment of a potentially difficult airway.

Another important score for airway management also changed over time. Intubation conditions were increasingly rarely rated as very good in terms of a Cormack-Lehane score I. At the same time, the frequency of the Cormack-Lehane scores II-IV remained unchanged. One explanation could be the simultaneous increase in the frequency of "not evaluable" Cormack-Lehane scores. This could be due to the fact that over time, supraglottic airway devices have been increasingly used, and since this airway device does not allow a view of the glottis, the Cormack-Lehane score could not be determined. At the same time, videolaryngoscopy was increasingly used in our department. Although the current German guidelines recommend the evaluation of the Cormack-Lehane score also for videolaryngoscopy [17] due to its simplicity and lack of an accepted alternative, many anaesthetists are under the impression that the Cormack-Lehane score is not suitable for videolaryngoscopy. This could be another reason for the prevalence of the "not evaluable" score. In the future, more emphasis should be placed on an adoption of a more suitable alternative of the Cormack-Lehane score during

videolaryngoscopy (e.g., digital videolaryngoscopic images within health records) to ensure patient safety by providing reliable a clearly understandable documentation of a potentially severe airway [18].

Compared to the other specialities, fibreoptic intubation has been used much more frequently in otorhinolaryngology and maxillofacial surgery. Due to the underlying pathologies, patients in the otorhinolaryngology and maxillofacial surgery are more likely to have a difficult airway. Thus, it was observed that in maxillofacial surgery, the use of fibreoptic intubation doubled between the years 2005 and 2010 and remained relatively stable thereafter. This can be attributed to fluctuating numbers of major cancer surgery in the department.

Although one might assume that videolaryngoscopy would replace fibreoptic intubation over the years, surprisingly no decline in fibreoptic intubation was observed [19]. This could possibly be due to the fact that fibreoptic intubation is considered an important procedure in our hospital. Every anaesthetist must be well trained for fibreoptic intubation, as according to guidelines, videolaryngoscopy is not an adequate substitute for the management of an anticipated difficult airway [20,21]. Therefore, the indication for fibreoptic intubation is generously given in our clinic to ensure adequate training, which is reflected in the increase of fibreoptic intubation use over time. This is in accordance with the recent Difficult Airway Society recommendations for awake tracheal intubation in adults [22].

We acknowledge that our data analysis has certain limitations. First, it does not allow for correlation of the type of airway management procedure with complications or the outcome of the patient, as this information was not documented in the DAQ. It would be interesting to align trends with patient outcome to allow for an assessment of the efficacy of airway management techniques and devices. Second, we have no data on videolaryngoscopy, which could help to conclusively evaluate the observed changes in fibreoptic intubation and the Cormack-Lehane scores. Last, although a large number of patients was evaluated, they were all treated in a single centre and hence the trends may differ in other clinics according to local preferences. University Medical Centre Mainz hosts a large department of oral and maxillofacial surgery, as well as a large department of otorhinolaryngology. We therefore treat an exceptional number of patients with a difficult airway. Patients with large tumors and abscesses in the airway and patients with rare syndromal and anatomical peculiarities belong to our complex patient population. Due to this condition, the results of this analysis cannot be transferred to other health care institutions. Consequently, external validity is therefore limited.

Concerning the assessment of data quality, it can be stated that objectively evaluated data like patient demographics and frequency of airway management techniques are unlikely to contain misclassification errors whereas evaluation of patient characteristics has limited reliability, and thus has potentially higher probability of misclassification. In particular data extraction procedures from the DAQ were automated to avoid errors from manual repetitive tasks.

In summary, a retrospective analysis from our in-house data acquisition revealed the pattern and changes in airway management over the last 16 years in a German university hospital. During this period, 340,748 documented airway procedures were performed, with oral intubation being the most frequent technique, followed by the use of supraglottic airway devices and fibreoptic intubations. Of note, there was a significant upward trend in the use of supraglottic airway devices, with an increase in the use of second-generation masks due to their improved characteristics compared to first-generation masks, while a decrease in oral intubations was observed.

Guidelines are intended to ensure a high degree of patient safety and at the same time recommend a corridor of action for the treating anesthesiologist. It is useful and necessary to adapt the respective guidelines to one's own everyday life and to the existing circumstances.

The clinical (regional or local) context should be considered an important factor for enhancing adherence to the recommendations by physicians and health systems [23]. Developing and updating practice-based guidelines to improve airway management and patient safety requires identification of changes and trends resulting from such retrospective, long-term data evaluations, but should ideally be correlated with clinical outcomes in future studies.

The establishment of an international registry with data on current airway management practice and detailed description of associated serious complications would be a milestone in airway management.

## Supporting information

**S1 Fig. Screenshot of the internal data acquisition system (DAQ) showing patient and airway evaluation.** Purple boxes are mandatory fields. English translation of relevant text elements (red box): Hirntod = brain death, Eingriffsart = type of surgery, Wahleingriff = elective, dringlicher Eingriff = urgent, Noteingriff = emergency, Kopfreklination = neck reclination, unauffällig = normal, fraglich = questionable, pathologisch = pathological, nicht beurteilbar = not evaluated, nicht untersucht = not analysed, Kinn/Hals = thyromental distance.
(GIF)

**S2 Fig. Screenshot of the internal data acquisition system (DAQ) showing anaesthesia technique, administered medication, and utilised airway management technique.** English translation of relevant text elements (red box): Luftweg = airway management technique.
(GIF)

**S1 Data.**
(CSV)

**S2 Data.**
(CSV)

**S3 Data.**
(CSV)

**S4 Data.**
(CSV)

## Acknowledgments

This study was supported by the Mainz Research School of Translational Biomedicine (TransMed) of the University Medical Centre, Johannes Gutenberg-University Mainz, Germany. Data shown in this manuscript are part of the professorial dissertation (Habilitation) of RH and NP presented to the Faculty of Medicine of the Johannes Gutenberg-University Mainz. Finally, we would like to thank all anaesthesiologists in the Department of Anaesthesiology, University Medical Centre Mainz, for their support and careful completion of our internal data acquisition system.

## Author Contributions

**Conceptualization:** Regina Hummel, Hans-Jürgen Baldering, Kristin Engelhard, Eva Wittenmeier, Katharina Epp, Nina Pirlich.

**Data curation:** Regina Hummel, Hans-Jürgen Baldering, Kristin Engelhard, Nina Pirlich.

**Formal analysis:** Regina Hummel, Daniel Wollschläger, Hans-Jürgen Baldering, Eva Wittenmeier, Katharina Epp, Nina Pirlich.

**Investigation:** Regina Hummel, Hans-Jürgen Baldering, Eva Wittenmeier, Nina Pirlich.

**Methodology:** Regina Hummel, Daniel Wollschläger, Hans-Jürgen Baldering, Kristin Engelhard, Nina Pirlich.

**Project administration:** Regina Hummel, Nina Pirlich.

**Resources:** Regina Hummel, Kristin Engelhard, Nina Pirlich.

**Software:** Regina Hummel, Daniel Wollschläger, Hans-Jürgen Baldering, Katharina Epp, Nina Pirlich.

**Supervision:** Regina Hummel, Daniel Wollschläger, Hans-Jürgen Baldering, Kristin Engelhard, Eva Wittenmeier, Katharina Epp, Nina Pirlich.

**Validation:** Regina Hummel, Daniel Wollschläger, Hans-Jürgen Baldering, Kristin Engelhard, Eva Wittenmeier, Nina Pirlich.

**Visualization:** Regina Hummel, Daniel Wollschläger, Hans-Jürgen Baldering, Eva Wittenmeier, Katharina Epp, Nina Pirlich.

**Writing – original draft:** Regina Hummel, Nina Pirlich.

**Writing – review & editing:** Regina Hummel, Daniel Wollschläger, Hans-Jürgen Baldering, Kristin Engelhard, Eva Wittenmeier, Katharina Epp, Nina Pirlich.

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
