## [Decision Letter · Decision Letter 0]

30 May 2022

PONE-D-22-11287Big data: Airway management at a university hospital over 16 years; a retrospective analysisPLOS ONE

Dear Dr. Pirlich,

Thank you for submitting your manuscript to PLOS ONE. After careful consideration, we feel that it has merit but does not fully meet PLOS ONE’s publication criteria as it currently stands. Therefore, we invite you to submit a revised version of the manuscript that addresses the points raised during the review process.

We look forward to receiving your revised manuscript.

Kind regards,

Marcus Tolentino Silva

Academic Editor

PLOS ONE

Journal Requirements:

a) Did participants provide their written or verbal informed consent to participate in this study? b) If consent was verbal, please explain i) why written consent was not obtained, ii) how you documented participant consent, and iii) whether the ethics committees/IRB approved this consent procedure.

Reviewers' comments:

Reviewer's Responses to Questions

**Comments to the Author**

1. Is the manuscript technically sound, and do the data support the conclusions?

Reviewer #1: Yes

Reviewer #2: Yes

Reviewer #3: Partly

2. Has the statistical analysis been performed appropriately and rigorously? 

Reviewer #1: Yes

Reviewer #2: Yes

Reviewer #3: Yes

3. Have the authors made all data underlying the findings in their manuscript fully available?

Reviewer #1: No

Reviewer #2: Yes

Reviewer #3: No

4. Is the manuscript presented in an intelligible fashion and written in standard English?

Reviewer #1: Yes

Reviewer #2: Yes

Reviewer #3: Yes

5. Review Comments to the Author

Reviewer #1: Peer review PONE

Big data: Airway management at a university hospital over 16 years; a retrospective analysis

In the age of ever-increasing knowledge regarding airway management, to know details and trends in time about the procedures, techniques and uses of the approaches remains very useful. I appreciate the opportunity to carefully read and comment regarding the submission “Big data: Airway management at a university hospital over 16 years; a retrospective analysis”. The idea and the availability of data to examining the trends over time in such a big number of procedures is commendable and with major clinical application.

Major comments,

• Most analytic goals for studies can be viewed as either seeking evidence of causality or prediction. From my early view, I feel you really present a descriptive analysis, and this is wonderful. I really congratulate about that. Many people think if they are not doing exploratory analysis o some related are not doing good science and that is a big mistake (https://doi.org/10.1038/s41416-020-1019-z). Research studies that focus on describing a population of interest are essential building blocks for both causal and predictive frameworks, and do not typically require control for additional variables. To understand the purpose of a study (i.e., descriptive, causal, or predictive), it is vital that the goals of the research be clearly explained. Please consider adjusting your purposes section of the abstract to become it more transparent and clearer (see below).

• Please, do not emphasize in the whole text any analysis on the basis of confounding. Here, I can see a great use of descriptive epidemiology beyond any causal purpose.

Minor comments per section,

Abstract

• I find the abstract quite difficult to understand. First, the purpose does not clarify in detail the aims of this study limiting itself to the need of data in this area. Please, consider clarifying the aims of this study at the end of the first section: “purpose”.

• Let’s consider adding to the methods section if you are including only adults, children, pregnant patients, etc.

• In the results section, please avoid the use of only significance testing during reporting of results. Quote “Over time, second generation laryngeal masks were used more frequently than first-generation laryngeal masks (p < 0.001)” can be adjusted by adding the degree of reduction over time, or at least over two extreme years.

• The conclusion seems to be a repetition of the results. Please conclude by following your results, perhaps regarding the trends directly and the reduction on the use of certain techniques over time.

Introduction

• It is clear the aims of your study after reading the full introduction.

Methods

• It is quite impressive that in such amount of data you do not have missing values. Methods section describes very well the process of data collection from the system, but it does not mention at all any relevant information for potential information bias from that system, how did you address that? is there any potential variable to be missed or at least biased?

• Following the previous comment, STROBE request information about how missing data or potential biases were addressed. It is quite relevant to provide information about to enhance completeness of you report. Please add relevant information.

• Results section mention “For 15,756 anaesthesia patients (4.6%) the type of procedure was not evaluated.”. Please, would you clarify why? Are those patients considered as missing values from the analysis?

• I may understand you used all data from your collection system. Anyway, it is relevant ot provide information to reader about sample size. It was not calculated so that should be stated and mentioned. Explain how the study size was arrived at?

Results

• Study cohort and characteristics. Always it is a good idea to provide a Flowchart regarding data sources and inclusion/exclusion criteria used. Indeed, completeness checklist includes a flowchart diagram as initial part of results section. Please, consider adding that flowchart of patient selection and inclusion. Report numbers of individuals at each stage of study—eg. numbers potentially eligible, examined for eligibility, confirmed eligible, included in the study, completing follow-up, and analyzed

• Figures 1-6 are very well presented and attractive. Please add all relevant information to each Figure to the reader, including colors.

Discussion

• While I agree with this point “To our knowledge, there are only few studies with a comparable large data set of …”, I would like to add also this fact reduces external validity of your results. I do not see any section of the discussion taking about the type of patients in terms of complexity you receive in your center. Are those patients reflecting the common clinical practice in other health care systems? In what extent? Please, expand about that and discuss the generalizability (external validity) of the study results

• Discussion is very well conducted and interesting enough. To me, as a reader it was very nice to explore.

• You mention clear limitations. However, measurement error was not included explicitly. Quite complex in retrospective datasets. Any potential to misclassify exposure, outcome or included covariates. Any consideration about steps taken to ensure accurate measurement (double extraction, double check). Thank you for expanding on these details.

• You mention several times in the text the importance of this data to clinical practice guidelines. Quote “…and serve as a basis for future adaptation of airway management guidelines.”. Do your consider people in your hospital or perhaps country is following current guidelines in airway management and that is reflected in your data? I would like to suggest citing and mention that current guidelines in airway management seems to be well developed but poorly able to be adapted to clinical practice (CPGs show low applicability of their recommendations to real clinical practice) (https://doi.org/10.1097/eja.0000000000001195 ). Please, could you expand on this topic perhaps a final paragraph.

Reviewer #2: This is an interesting well written manuscript that in my opinion is adequate for publication after minor revision.

I only have two comments:

In limitations it should be mentioned that data comes from single hospital, so the external validity is limited.

Study period goes until just before the COVID 19 pandemic started. Some comment on the effect of the pandemic on the results could be interesting

Reviewer #3: I would like to commend the authors for looking at 16 years of data and in excess of 340,000 airway management procedures. This is big data.

Specific comments:

Page 5 line 50-64: This portion may not be immediately relevant to the study and can be shortened or summarized.

Page 7 line 93: Cormack and Lehane grading may not be airway evaluation data but rather intubation grading data.

Page 10 line 150: Data from 414,843 patients were analyzed but only 340748 patients were included. Could the authors explain in more detail why this was so?

General comments about the intrinsic limitations of the study as presented by the authors:

1. 16 years ago airway management may be different. Use of 2nd generation supraglottic airway devices and video laryngoscopes more widespread in the last decade. I wonder if limiting the period to 10 years may be more relevant to practicing Anaesthesiologists.

2. Use of laryngeal mask only 27% in 2020. That figure seems a little low and is not in keeping with international data and my own experience. It should be closer to 50%, of course depending on the type of cases that are performed.

3. One main finding is that laryngeal mask use has increased over time. This has been very well reported in the literature. It would be interesting to know if the use of video laryngoscopes have likewise increased over time.

4. A more interesting slant may also be a subgroup analysis of the use of fibreoptic intubation, e.g. indications, complications, failure rate, etc.

5. There are generally too many figures. These should be reduced.

6. Conclusions by the author are not novel and many (if not all) of these have been espoused in other prior publications or are intuitive.

Minor comment:

1. Consider using the term supraglottic airway device instead of laryngeal mask (which is a brand name). Consider detailing the types of supraglottic airway devices used during the 16 year period (e.g. Ambu AuraGain, LMA Protector, Igel, etc)

6. PLOS authors have the option to publish the peer review history of their article (what does this mean?). If published, this will include your full peer review and any attached files.

Reviewer #1: **Yes: **Jose Andres Calvache

Reviewer #2: No

Reviewer #3: No

---

## [Author Response · Author response to Decision Letter 0]

19 Jul 2022

Please find the response in the uploaded file "Response to Reviewers"

---

## [Decision Letter · Decision Letter 1]

11 Aug 2022

Big data: Airway management at a university hospital over 16 years; a retrospective analysis

PONE-D-22-11287R1

Dear Dr. Pirlich,

We’re pleased to inform you that your manuscript has been judged scientifically suitable for publication and will be formally accepted for publication once it meets all outstanding technical requirements.

Kind regards,

Marcus Tolentino Silva

Academic Editor

PLOS ONE

Additional Editor Comments (optional):

Reviewers' comments:

Reviewer's Responses to Questions

**Comments to the Author**

1. If the authors have adequately addressed your comments raised in a previous round of review and you feel that this manuscript is now acceptable for publication, you may indicate that here to bypass the “Comments to the Author” section, enter your conflict of interest statement in the “Confidential to Editor” section, and submit your "Accept" recommendation.

Reviewer #1: All comments have been addressed

Reviewer #2: All comments have been addressed

Reviewer #3: All comments have been addressed

2. Is the manuscript technically sound, and do the data support the conclusions?

Reviewer #1: Yes

Reviewer #2: Yes

Reviewer #3: Partly

3. Has the statistical analysis been performed appropriately and rigorously? 

Reviewer #1: Yes

Reviewer #2: Yes

Reviewer #3: I Don't Know

4. Have the authors made all data underlying the findings in their manuscript fully available?

Reviewer #1: Yes

Reviewer #2: Yes

Reviewer #3: Yes

5. Is the manuscript presented in an intelligible fashion and written in standard English?

Reviewer #1: Yes

Reviewer #2: Yes

Reviewer #3: Yes

6. Review Comments to the Author

Reviewer #1: Thank you for addressing all comments I made. This initiative is a good approach to big data information in anesthesiology. Congrats for this great manuscript !

Reviewer #2: The authors have answered all the comments to the reviewers. So in my oppinion the manuscript is ready for publication

Reviewer #3: I would like to thank the authors for responding to the comments. The revised manuscript has addressed some of the inaccuracies from the original manuscript and highlighted relevant limitations.

This is a study analyzing more than a decade of airway management experience in a tertiary hospital in Europe.

The findings and final conclusion, however, are not novel. It points towards something which is well known from the existing literature, vis-a-vis the increase in use of supraglottic airway devices and decrease in tracheal intubations. This is especially so with the proliferation of second generation supraglottic airway devices.

7. PLOS authors have the option to publish the peer review history of their article (what does this mean?). If published, this will include your full peer review and any attached files.

Reviewer #1: **Yes: **Jose A. Calvache

Reviewer #2: No

Reviewer #3: No

---

## [Editor Report · Acceptance letter]

12 Sep 2022

PONE-D-22-11287R1 

Big data: Airway management at a university hospital over 16 years; a retrospective analysis 

Dear Dr. Pirlich:

I'm pleased to inform you that your manuscript has been deemed suitable for publication in PLOS ONE. Congratulations! Your manuscript is now with our production department. 

Kind regards, 

on behalf of

Dr. Marcus Tolentino Silva 

Academic Editor

PLOS ONE